# Increased Angiogenesis and Lymphangiogenesis in Adenomyosis Visualized by Multiplex Immunohistochemistry

**DOI:** 10.3390/ijms23158434

**Published:** 2022-07-29

**Authors:** Marissa J. Harmsen, Arda Arduç, Maaike C. G. Bleeker, Lynda J. M. Juffermans, Arjan W. Griffioen, Ekaterina S. Jordanova, Judith A. F. Huirne

**Affiliations:** 1Department of Obstetrics and Gynaecology, Amsterdam UMC Location Vrije Universiteit Amsterdam, De Boelelaan 1117, 1081 HV Amsterdam, The Netherlands; a.arduc@amsterdamumc.nl (A.A.); ljm.juffermans@amsterdamumc.nl (L.J.M.J.); j.huirne@amsterdamumc.nl (J.A.F.H.); 2Amsterdam Reproduction and Development Research Institute, 1105 AZ Amsterdam, The Netherlands; e.jordanova@amsterdamumc.nl; 3Department of Pathology, Amsterdam UMC Location Vrije Universiteit Amsterdam, De Boelelaan 1117, 1081 HV Amsterdam, The Netherlands; mcg.bleeker@amsterdamumc.nl; 4Cancer Center Amsterdam, De Boelelaan 1117, 1081 HV Amsterdam, The Netherlands; a.griffioen@amsterdamumc.nl; 5Angiogenesis Laboratory, Department of Medical Oncology, Amsterdam UMC Location Vrije Universiteit Amsterdam, De Boelelaan 1117, 1081 HV Amsterdam, The Netherlands; 6Center for Gynecologic Oncology Amsterdam, Amsterdam UMC Location Vrije Universiteit Amsterdam, 1081 HV Amsterdam, The Netherlands; 7Department of Urology, The Netherlands Cancer Institute, 1066 CX Amsterdam, The Netherlands

**Keywords:** adenomyosis, angiogenesis, lymphangiogenesis, ectopic endometrium, histology

## Abstract

There is evidence for increased angiogenesis in the (ectopic) endometrium of adenomyosis patients under the influence of vascular endothelial growth factor (VEGF). VEGF stimulates both angiogenesis and lymph-angiogenesis. However, information on lymph vessels in the (ectopic) endometrium of adenomyosis patients is lacking. In this retrospective matched case-control study, multiplex immunohistochemistry was performed on thirty-eight paraffin embedded specimens from premenopausal women who had undergone a hysterectomy at the Amsterdam UMC between 2001 and 2018 to investigate the evidence for (lymph) angiogenesis in the (ectopic) endometrium or myometrium of patients with adenomyosis versus controls with unrelated pathologies. Baseline characteristics of both groups were comparable. In the proliferative phase, the blood and lymph vessel densities were, respectively, higher in the ectopic and eutopic endometrium of patients with adenomyosis than in the endometrium of controls. The relative number of blood vessels without α-smooth muscle actinin (α SMA) was higher in the eutopic and ectopic endometrium of adenomyosis patients versus controls. The level of VEGF staining intensity was highest in the myometrium but did not differ between patients with adenomyosis or controls. The results indicate increased angiogenesis and lymphangiogenesis in the (ectopic) endometrium affected by adenomyosis. The clinical relevance of our findings should be confirmed in prospective clinical studies.

## 1. Introduction

Adenomyosis is a benign uterine condition where endometrium glands and stroma that normally line the uterine cavity (eutopic endometrium) are present in the myometrium (ectopic endometrium) [1,2]. To understand the pathophysiology of adenomyosis and the associated symptoms of abnormal uterine bleeding and subfertility, it is vital to gain more insight into the morphology of adenomyosis tissue in the uterus and the difference with normal endo-myometrial tissue.

There is evidence for increased angiogenesis in the endometrium of adenomyosis patients and there are indications that the symptoms of adenomyosis patients might be related to the presence of ongoing angiogenesis in the eutopic and ectopic endometrium [3]. Angiogenesis is the growth of new blood vessels from preexisting ones [4]. A theory for the involvement of angiogenesis in the pathogenesis of adenomyosis is that the repetitive nature of the contractions of the menstrual cycle causes tissue injury to the junctional zone, invagination of the endometrium in the myometrium, and leads to angiogenesis in the process of repair and establishment of endometrial tissue at ectopic sites [5,6]. Previous studies have suggested that angiogenesis may play a crucial role in the pathophysiology of adenomyosis-related abnormal uterine bleeding [3,7,8,9,10,11,12]. In addition, there is indirect evidence for the association between angiogenesis and subfertility in adenomyosis patients [3]. Members of the vascular endothelial cell growth factor (VEGF) family stimulate both angiogenesis and lymphangiogenesis [13], and an increase in the density of lymph vessels has been found in the stroma of endometrial lesions [14,15]. Distorted lymphatic drainage might also be a factor in the observed subfertility in adenomyosis patients. This theory is based on the observation that the clinical pregnancy rate after assisted reproductive technology (ART) was lower among patients with endometrial cavity fluid [16]. This fluid can consist of blood and endometrial secretions, and it might be present because of changes in lymph drainage.

Symptoms associated with adenomyosis, such as abnormal uterine bleeding and impaired fertility, might be due to vascular leakage during angiogenesis, increased interstitial fluid, and distorted lymphatic drainage in endometrial tissue. Therefore, the main objective of this study was to investigate whether there is evidence for changes in blood vessel and lymph vessel density. The relative number of vessels without α-SMA is an indication of how many capillaries and newly formed blood vessels a certain tissue contains and, hence, may be indicative of the rate of angiogenesis taking place. Finally, the level of VEGF expression was assessed, and all markers were stained for on the eutopic and ectopic endometrium and myometrium of patients with adenomyosis compared to the endometrium or myometrium in controls with unrelated pathologies.

## 2. Results

Baseline characteristics of the 19 patients diagnosed with adenomyosis and the 19 controls with unrelated pathology are presented in Table 1. The patients with adenomyosis and the controls were comparable in age, menstrual phase, and parity. Most adenomyosis patients (57.9%) and controls (73.3%) did not use any medication. A small proportion of adenomyosis patients (15.8%) and controls (26.3%) used nonsteroidal anti-inflammatory drugs (NSAIDs) on an incidental basis for dysmenorrhea in the 3 months prior to surgery. Ten adenomyosis patients (52.7%) had a previous cesarean section (Table 1). Out of the thirty-eight selected hysterectomy specimens, 12 adenomyosis patients and 17 controls had representative material in both the eutopic endometrium, ectopic endometrium (when applicable), and myometrium that could be analyzed. In the remaining adenomyosis patients and controls, eutopic endometrium samples could be analyzed from fourteen of the adenomyosis patients and seventeen of the controls. For four out of 19 adenomyosis patients and one out of 19 controls, there was no eutopic endometrium available, as it was not discernable by the pathologist in the available material. IHC staining failed for one adenomyosis patient and one control. Sixteen from the nineteen patients with adenomyosis had an ectopic endometrium to analyze, and others had no representative ectopic endometrium in the available material (2/19 adenomyosis patients) or IHC staining failed (1/19 adenomyosis patients). The myometrium could be analyzed in all selected specimens.

### 2.1. Multiplex Immunofluorescence

To analyze blood and lymph vessel density, structure and angiogenesis tissue specimens were stained for CD31, podoplanin, α-SMA, and VEGF in samples of the control eutopic endometrium, adenomyosis eutopic endometrium, and adenomyosis ectopic endometrium. Examples of the results of the multiplex staining results and the automated detection that was used for analyses are presented in Figure 1 and Figure 2, in which blood and lymph vessels are visualized in the endometrium stroma in between the endometrium glands, and in the myometrium, respectively. Representative images of the staining results of the individual fluorescent markers are presented in Figure 3 and Appendix A.

Tissue samples were divided according to the menstrual cycle into the secretory and proliferative phase. All results are presented per cycle phase. Overall, the blood vessel density in the eutopic endometrium was higher in the secretory phase than in the proliferative phase (*p* = 0.017; Figure 4). There was no difference in lymph vessel density between the proliferative and secretory phase (*p* = 0.804).

A subgroup analysis was also performed on patients that used medication—other than hormonal—that has an anti-angiogenic profile, such as NSAIDs or tranexamic acid, and those that did not use this medication (Figure 5). There was no significant difference in blood (*p* = 0.149) or lymph vessel density (*p* = 0.804) between these two groups. Therefore, these factors were not taken into account in further analyses.

### 2.2. Blood and Lymph Vessel Density

In both phases of the menstrual cycle, a higher blood vessel density (BVD) can be seen in the ectopic endometrium compared with the eutopic endometrium of adenomyosis patients (Figure 6A,B). This difference was significantly different in the proliferative phase (post hoc analysis *p* = 0.045) and borderline significant in the secretory phase (*p* = 0.051). The same pattern of increased BVD was observed when comparing ectopic BVD with the control endometrium in the proliferative phase (*p* = 0.016) and in the secretory phase (*p* = 0.121). In myometrial tissue, no difference was observed between adenomyosis patients and the control myometrium.

Instead of counting the number of CD31-stained areas (vessels), the percentage of CD31-positive area per CD10 or myometrium-stained area per MSI was also assessed, to verify that the outcome measure would yield comparable results. All differences found between %CD31-stained area per tissue type were comparable to the outcomes in vessel count per tissue type (Appendix A). In addition, this analysis showed a similar elevated %CD31-positive area in the ectopic endometrium compared with the eutopic endometrium, being significantly different in the proliferative phase (*p* = 0.006). There was no difference in %CD31-stained area in the myometrium between adenomyosis and control patients in either menstrual phase (secretory *p* = 0.957; proliferative *p* = 0.253).

In contrast with blood vessel density, which showed an increase in the ectopic endometrium, lymph vessel density showed an increase in the eutopic endometrium in the proliferative phase (Figure 6D) compared with the control endometrium (*p* = 0.045). There was no difference in LVD between adenomyosis and the control myometrium in either cycle phase (secretory *p* = 0.892; proliferative *p* = 1.00). In addition, for LVD, all results were comparable when the percentage of podoplanin-stained area was assessed per tissue type, with the only significant difference found between the percentage of podoplanin stained area in the eutopic vs. control endometrium (*p* = 0.045) (Appendix A).

### 2.3. Relative Number of Capillaries and Newly Formed Vessels

Both smooth muscle cells and pericytes can be visualized using anti-αSMA. Existing capillaries and newly formed vessels during angiogenesis have little to no pericytes surrounding the endothelial layer, whereas mature arterioles are covered with smooth muscle cells. The relative number of vessels without α-SMA is an indication of how many capillaries and newly formed blood vessels a certain tissue contains and hence may be indicative of the rate of angiogenesis taking place.

The relative number of mature blood vessels, determined by the ratio of vessels that were double-stained with CD31 and α-SMA to the total number of vessels (CD31 positive), was lower in both the eutopic and ectopic endometrium of adenomyosis patients compared with control patients. This difference was significant in the secretory phase for the ectopic endometrium vs. control (K-W *p* = 0.005, ectopic vs. control *p* = 0.003), and significant for both the eutopic and ectopic endometrium vs. control in the proliferative phase (K-W *p* = 0.014; eutopic vs. control endometrium *p* = 0.011, ectopic vs. control *p* = 0.016) (representative image examples in Figure 3) (Figure 7).

### 2.4. Immunohistochemical Score (IHS) of VEGF Expression per Tissue Type

To investigate the level of VEGF staining per tissue type in adenomyosis versus control patients, the immunohistochemical score (IHS) was calculated. Overall, the IHS of VEGF was highest in the myometrium (median VEGF IHS = 4, *n* = 37), followed by endometrium glands (median VEGF IHS = 3, *n* = 32), and lowest in the endometrium stroma (median VEGF IHS = 1, *n* = 32). There was no difference between the VEGF staining intensity per cycle phase, or between adenomyosis and control patients (Figure 8).

## 3. Discussion

The current study shows that there are signs of both angiogenesis and lymphangiogenesis in adenomyosis tissue, which are possibly fundamental in the pathogenesis of adenomyosis. We demonstrated that blood vessel density (BVD) was highest in the ectopic endometrium and increased compared with both the eutopic endometrium of patients with and without adenomyosis. The eutopic endometrium of adenomyosis patients had a higher relative number of blood vessels without αSMA, demonstrated by the significantly lower ratio of mature vessels to the total number of blood vessels compared with the control endometrium. Despite the increase in BVD in the ectopic endometrium, we found no increased levels of VEGF staining intensity in endometrial tissue, neither in eutopic nor in ectopic tissue. Lymph vessel density (LVD) was only higher in the eutopic endometrium of patients with adenomyosis compared with the endometrium of controls with unrelated pathology.

The increase in BVD was observed in both phases of the menstrual cycle. Yet only the difference in the proliferative phase was significantly different. However, the small number of patients could account for this, especially after dividing them into patients with samples taken during the secretory and samples taken during the proliferative phase. The increase in BVD is indicative of increased angiogenesis in endometrial tissue in adenomyotic patients.

While an increased BVD and expression of vascular endothelial growth factor (VEGF) in the ectopic endometrium have been reported by others [3,8,9,11,17,18,19], the results in the eutopic endometrium were ambiguous [10,20,21]. The findings in this study do not reveal a difference between the eutopic and control endometrium either, and an increased BVD was found in the ectopic endometrium only. In line with previous studies on BVD in patients with and without adenomyosis that took the menstrual phases into account, the BVD was overall higher in the secretory phase than in the proliferative phase [3,8,10]. We do see this for the pooled data (Figure 4), and only in the control endometrium, not in tissue from adenomyosis patients. Huang et al., noted that the expression of VEGF and BVD was higher in the secretory than in the early proliferative phase in both adenomyosis and control patients [8]. In a population of sixteen healthy women, the expression of VEGF receptors was highest in the secretory phase, while there was no correlation between VEGF receptor expression and stromal blood vessel density [22].

Angiogenesis and lymphangiogenesis are essential for the physiological regeneration of the endometrium following menstruation [23], as well as for decidualization of the endometrium in the secretory phase [24]. It has not been reported before whether there are also signs of angiogenesis in the myometrium of adenomyosis patients versus control patients. The assessment of the myometrium in this study resulted in the finding that VEGF expression is higher in this tissue type compared with endometrial tissue, but without a difference between patients with and without adenomyosis. Myometrial hypertrophy is a hallmark of adenomyosis, and in the affected myometrium, an abundance of myofibroblasts suggests that tissue injury and repair occur [25]. In this regard, one would expect an increased presence of VEGF in the myometrium of adenomyosis patients. The lack of difference between the VEGF staining intensity and the BVD in the adenomyosis myometrium versus control myometrium that was found in this study might be explained by the observation in a cross-sectional study by Liu et al., on ectopic endometrial tissue samples that, in cases of advanced adenomyosis, myometrial hypertrophy results in fibrosis and a concomitant decrease in vascular density [9]. However, this is a hypothetical assumption as the evaluation of fibrosis was beyond the scope of this study and could not be assessed in these samples.

The reason why angiogenesis might lead to abnormal uterine bleeding in adenomyosis patients is likely to be multifactorial. One theory is that pro-angiogenic factors, such as VEGF, stimulate proliferation of endothelial cells and potentiate microvascular hyperpermeability of uterine vessels under estrogen stimulation [26,27,28]. Subsequently, this hyperpermeability of the capillary vessels can lead to abnormal uterine bleeding (Middelkoop et al., 2022. Submitted). In the current study, we did not find evidence to support this theory, as the increased blood vessel density was merely found in the ectopic endometrium, which is unlikely to contribute to abnormal uterine bleeding and more likely to form myometrial cysts. However, in the eutopic endometrium of adenomyosis patients, the decreased presence of α-SMA surrounding blood vessels compared to the eutopic endometrium of patients without adenomyosis suggests another mechanism of angiogenesis-induced abnormal uterine bleeding [13,29]. The initiation of angiogenesis is accompanied by vascular leakage and increased interstitial fluid [30]. This theory is supported by the observed decreased levels of α-SMA expression reported in women with menorrhagia compared with controls [31]. Interestingly, a recent systematic review by Middelkoop et al., found that blood vessel density was generally not increased in patients with abnormal uterine bleeding, but that other morphological parameters, such as pericyte vessel wall coverage, were aberrant in the endometrium of this population [32], which is in line with this study. A lack of pericyte coverage and an increase in the number of blood vessel wall gaps and defects have been associated with abnormal uterine bleeding in the endometrium [32]. We hypothesize that the stimulus for this process derives from pro-angiogenic factors, which are increased in patients with adenomyosis [3].

The significant difference between LVD in the eutopic endometrium of adenomyosis patients versus the control endometrium is in line with previous research. Cho et al., showed that patients with adenomyosis had a higher LVD than patients with adenomyosis who were treated with a levonorgestrel-releasing intra-uterine device (LNG-IUD) or patients with cervical intraepithelial neoplasia that served as controls [33]. In addition, women with endometriosis LVD were found to have an increased eutopic endometrium, highest during menstruation [34]. As endometriosis and adenomyosis often co-exist and may share a common pathophysiology, the lymphatic spread of endometrial cells might be of importance in adenomyosis as well [35]. Interestingly, evidence for this theory was reported in five cases where endometrioid glandular and stromal elements were found in venous and lymphatic spaces within the myometrium [36]. A higher LVD in the eutopic endometrium might indicate a facilitated uptake of endometrium cells in the lymphatic vessels, which could be distributed throughout the underlying myometrium [15]. In addition to its role in the pathophysiology of adenomyosis, the increased LVD that was found in the eutopic endometrium of adenomyosis patients supports our hypothesis that lymphangiogenesis might be a factor in the observed subfertility in adenomyosis patients.

### 3.1. Strengths and Limitations

Currently, there are no studies that examined the role of angiogenesis or lymphangiogenesis in adenomyosis with multiplex immunohistochemistry. This new technique allowed the simultaneous visualization of the various markers of interest in the morphological context of the same tissue sample. This allowed for a detailed distinction between areas of endometrium, stained by CD10, and myometrium, stained by α-SMA, and a vessel count in each tissue type. Another strength was the application of automated analysis of the different fluorescence signals, which prevented selection bias and counting errors. A limitation of the applied technique is that the analysis is dependent on the selected areas of interest (MSI) instead of whole slide analysis, which would yield a more accurate estimate of vessel density [37]. While some studies chose the areas of interest randomly, others used a ‘hotspot’ method and chose the area of tissue with the highest number of vessels [9,33]. Another limitation of this study is the limited number of patient samples and the retrospective character. The differentiation between the menstrual cycle phases further decreased the number of samples that could be compared. The small patient sample size gives reason to interpret the results of this study with caution. The statistical power of such small patient samples is low, and both the positive as well as the negative findings might be underpowered to draw valid conclusions. However, using multiplex IHC can be regarded as a more specific and detailed method to analyze the markers of interest in the correct tissue type (i.e., vessel density only in the ectopic myometrium). Thus, the methodology of this study could be repeated in a larger, preferably prospectively collected patient sample to assess the reproducibility of the results. However, although adenomyosis is nowadays better recognized in the outpatient setting, and consequently more hysterectomies are performed for this reason, the difficulty would be to find appropriate control patients, as hysterectomies are not regularly performed in premenopausal women with prolapse symptoms without endo-myometrial pathology. In addition, in this study, the fact that the control patients required a hysterectomy to alleviate their symptoms might indicate that there were aberrations in angiogenesis and/or lymphangiogenesis in these uteri as well. Nonetheless, it can be considered a strength of this study that all controls were matched for age and parity with the cases to strive for a homogeneous group.

### 3.2. Wider Implications and Future Research

As adenomyosis is associated with AUB and impaired fertility, it would be interesting to investigate whether the increased presence of angiogenesis and lymphangiogenesis that was found in this study is a pathogenic factor in AUB or infertility, possibly negatively affecting the endometrial receptivity. Although it has become clear that there are signs of increased angiogenesis and lymphangiogenesis in the eutopic and ectopic endometrium of patients with adenomyosis, it remains unclear what the clinical consequences of an increased blood or lymph vessel density are. The current study should be replicated in a larger, prospective cohort of patients with adenomyosis to be able to link the clinical symptoms to the possible presence of angiogenesis and/or lymphangiogenesis. The exact location and extent of the adenomyotic lesions should be recorded to be able to assess the severity of adenomyosis in relation to the outcome. In addition, the exact day of the menstrual cycle should be recorded to further investigate and characterize the association between the menstrual cycle phase and angiogenesis and lymphangiogenesis parameters. As all studies performed so far have used hysterectomy specimens, the patients that opt against a hysterectomy, possibly because of a wish to conceive, could not participate. A specified imaging technique to show increased vessel density on the level of the microcirculation would be required to assess the correlation between vascularity and clinical symptoms, and this is highly needed in this field of research.

## 4. Materials and Methods

### 4.1. Tissue Samples

For this retrospective matched case-control study, paraffin-embedded samples were obtained from thirty-eight premenopausal women who had undergone a hysterectomy at the Amsterdam UMC, location VUmc, Amsterdam, Netherlands, between 2001 and 2018. The tissues were obtained from the Biobank of the Amsterdam UMC, location VUmc after ethical approval from the medical ethical committee. The cases were nineteen patients with histology-confirmed adenomyosis, who were selected from pathology reports that reported ‘endometriosis interna’ or ‘diffuse adenomyosis’. There were no cases of focal adenomyosis. The patients with adenomyosis underwent a hysterectomy for abnormal uterine bleeding (*n* = 15, 78.9%), dysmenorrhea (*n* = 3, 15.8%), or cervical dysplasia (*n* = 1, 5.3%). These were matched for age and parity with nineteen control patients who had a hysterectomy for pelvic organ prolapse (*n* = 11, 57.9%), dysmenorrhea (*n* = 3, 15.8%), cervical dysplasia (*n* = 3, 15.8%), an ovarian cyst (*n* = 1, 5.3%), or abnormal uterine bleeding (*n* = 1, 5.3%), but histologically no adenomyosis, endometriosis, or uterine fibroids (Table 1). Patients with a previous cesarean section were not eligible as controls, as a cesarean scar defect and adenomyosis often coincide. None of these patients used hormonal therapy (gonadotropin-releasing hormones or hormonal or intrauterine contraception) in the 3 months prior to surgery. For each case of adenomyosis, a pathologist specialized in gynecological pathology (MB) selected representative slides of the eutopic endometrium from the corpus uteri (not in proximity of a potential cesarean section scar) lining the uterine cavity and overlying the ectopic endometrium present inside the myometrium. All samples of the eutopic and ectopic endometrium were derived from the same slide. The definition of adenomyosis was the presence of myometrial invasion > 2.5 mm by ectopic endometrium tissue. From each control patient, a representative slide of the endometrium and myometrium was selected, after confirming there was no other pathology present in the endometrium or myometrium. Both case and control slides were classified by the pathologist according to the phase in the menstrual cycle as either ‘proliferative’ or ‘secretory’ based on the morphological appearance of the endometrium, as described by Noyes et al. [38]. In short, the proliferative endometrium was characterized by glands with narrow lumen, pseudostratification, and the presence of mitotic figures, whereas the secretory endometrium was, depending on the phase of the secretion, characterized by subnuclear vacuoles, signs of secretion, or the presence of spiral arterioles with decidualization and inflammatory cells in the stroma. In the situation where no representative eutopic or ectopic endometrium was discernable in the available material, information on the cycle phase or pathology was retrieved from the initial pathology report.

### 4.2. Multiplex Immunohistochemistry

For this study, 5 µm sections of paraffin-embedded tissues were cut on adhesive glass slides. Human tonsil sections were used as positive controls. As negative controls, sections from the dataset were incubated with only the secondary antibody. The OpalTM 7-color Fluorescent IHC67y kit from Akoya Biosciences was used for antigen visualization as previously described [39,40]. The details of the multiplex panel can be found in Appendix A. The slides underwent deparaffinization with xylene and rehydration in ethanol. Blocking was performed for endogenous peroxidase using 0.3% H_2_O_2_/methanol for 20 min in the dark and an extra fixation step was included with 10% neutral-buffered formalin for 20 min (Leica Biosystems, Wetzlar, Germany), followed by 2 × 2 min in Milli-Q water and 2 min in 0.05% Tween20 in Tris-buffered saline (TBST). Antigen retrieval was carried out by heating the tissues in 0.05% ProClin300/Tris-EDTA (pH 9.0) antigen retrieval buffer in an 800 W standard microwave at 100% power until boiling point, followed by 15 min at 30% power. Slides were cooled down at room temperature (RT) and washed in Milli-Q water and in TBST with agitation at 30 rounds per minute (rpm) on a shaker. For blocking, slides were incubated with antibody diluent (Immunologic, Duiven, The Netherlands) in a humidified chamber for 10 min at RT with agitation at 30 rpm. After that, slides were incubated with antibody diluted overnight at RT. The following primary antibodies were used: mouse monoclonal anti-α-smooth muscle actin (α-SMA) (1A4; 1:1500; DAKO, Santa Clara, CA, USA) as a marker for (vascular) smooth muscle cells, mouse monoclonal anti-CD31 (JC70A; 1:50; DAKO) as marker for endothelial cells, human monoclonal anti-podoplanin (D2-40; 1:50; BIO-RAD, Hercules, CA, USA) as marker for lymphatic endothelium, rabbit polyclonal anti-human VEGF A-20 (sc-152; 1:500; Santa Cruz Biotechnology, Dallas, TX, USA) as a proangiogenic factor, and mouse monoclonal anti-CD10 (56C6; 1:50; Sanbio, Tokyo, Japan) as an endometrial stroma marker. For each primary antibody, all mentioned steps were repeated. Next, slides were washed 3× 2 min in TBST at RT and with agitation at 30 rpm and were subsequently incubated with Secondary Antibody Working solution (goat-α-rabbit/mouse) for 15 min at RT and at 30 rpm. Slides were washed 3 × 2 min in TBST at RT and 30 rpm and incubated with Opal fluorochromes (Opal520, Opal570, Opal650, Opal690, Opal540, and Opal620) diluted 1:150 in amplification diluent (all provided by the OPAL 7-color fluorescence IHC Kit) for 10 min at RT and 30 rpm. Slides were then washed 3 × 2 min in TBST at RT and 30 rpm. The slides underwent microwave treatment with AR6 buffer, were cooled down in ice-water, and were washed for 2 min in Milli-Q water and for 2 min in TSBT at 30 rpm. Nuclear counterstain DAPI working solution (provided by the OPAL 7-color fluorescence IHC Kit) was applied for 5 min in a humidity chamber, and the slides were washed again in TSBT and Milli-Q water at 30 rpm. Finally, the slides were mounted under coverslips with Diamond antifade mounting medium (Life Technologies, Carlsbad, CA, USA).

Images of all sections were obtained with a multiparameter scanning microscope, the Vectra Polaris. A series of five multispectral image (MSI) areas of interest were selected from representative parts of the eutopic endometrium, ectopic endometrium, and myometrium, excluding the vascular arcade, per tissue sample using Akoya’s Phenochart™ whole slide viewer (Figure 9). All magnifications were made ×200 with each field representing 0.9 × 0.7 mm. The MSI images were generated and viewed in PerkinElmer’s inForm^®^ (Waltham, MA, USA) advanced image analysis software. The surface area (mm^2^) of the different tissue types was quantified and used to calculate the count of blood or lymph vessels per surface area (blood and lymph vessel density, respectively) using automated analysis in NIS Elements (NIKON Instruments, Inc., Melville, NY, USA). All endothelial cells or cell clusters were taken into account for microvessel counting, as reported previously [41]. Blood vessel density (BVD) and lymph vessel density (LVD) were defined as the mean vessel count obtained per mm^2^ surface area of endometrium (CD10 stained area) or myometrium (α-SMA-stained area) [42]. Instead of the count of CD31-stained areas (vessels), the percentage of the CD10 or myometrium-stained area per MSI that was CD31-stained was also assessed, to verify that this outcome measure should yield comparable results. The blood vessels that presented double immunostaining of CD31 and α-SMA, and thus represent blood vessels consisting of endothelium and vascular smooth muscle, were defined as mature blood vessels and were quantified as a proportion of the total number blood vessels in the eutopic and ectopic endometrium stroma, a method also applied by others [43]. This method cannot be applied in the myometrium, as all blood vessels are surrounded by smooth muscle cells (SMCs). The level of VEGF staining was assessed by a previously described immunohistochemical score (IHS) [8,44]. The IHS is calculated by multiplying an estimate of the percentage of immunoreactive cells (staining of 1–10% score 1; 11–50% score 2; 51–80% score 3; 81–100% score 4) with an estimate of the staining intensity (scale 0 to 4) [8]. The automated analysis was performed twice by the same investigator (MH) who was blinded for the sample number. A second investigator (AA) who was also blinded for the sample number repeated random samples from the analysis to verify the results. In case the analysis yielded different results between the investigators, the automated analysis was adjusted or repeated manually to reach consensus. Values were expressed as median numbers of vessels/mm^2^ in endometrium or myometrium.

### 4.3. Statistics

Statistical analyses were performed using SPSS version 22.0. Normality plots were made to test the normality of distribution. Continuous variables were expressed as median (range). Nonparametric tests were used to analyze baseline characteristics and the results of the measures of blood vessel density, lymph vessel density, ratio of mature to immature vessels, and VEGF staining intensity between both groups, including the Kruskal-Wallis test for the overall difference between multiple groups and Mann-Whitney U for comparisons between two groups. Results were considered statistically significant at the 5% level (two-tailed).

## 5. Conclusions

The findings in this study support the hypothesis that new and fragile blood and lymphatic vessels accompanied in the eutopic endometrium might be responsible for adenomyosis-associated symptoms, such as abnormal uterine bleeding and impaired fertility.

## Figures and Tables

**Figure 1 ijms-23-08434-f001:**
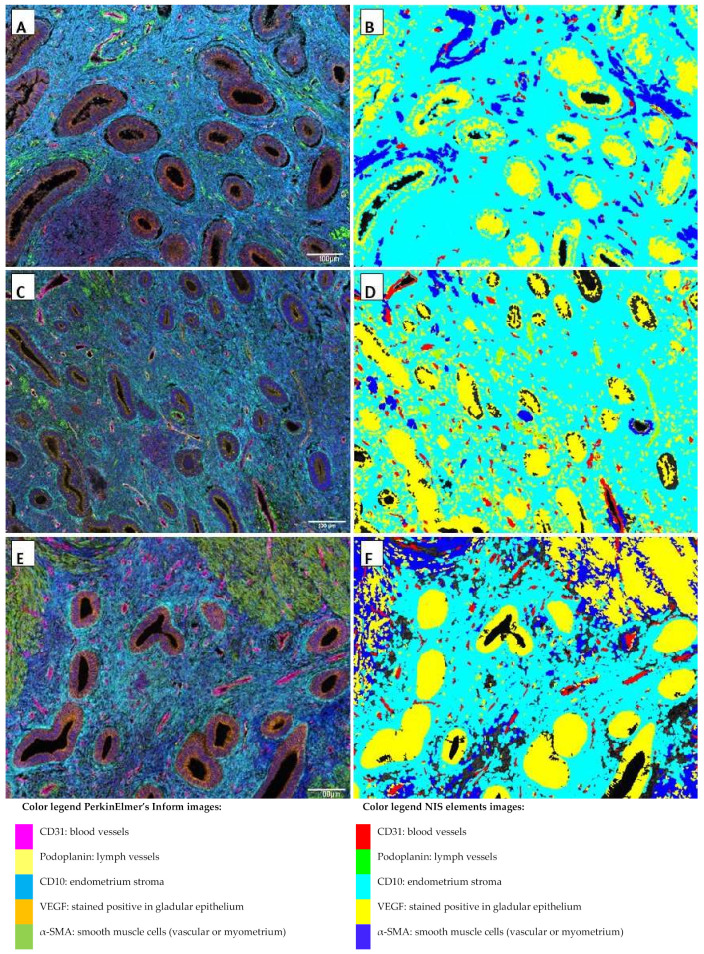
Multiplex images viewed in PerkinElmer’s inForm^®^ advanced image analysis software (**A**,**C**,**E**) and analysis of multiplex images in NIS Elements using binary layers for an automated analysis (**B**,**D**,**F**) of control endometrium (**A**,**B**), eutopic endometrium (**C**,**D**), and ectopic endometrium (**E**,**F**). All magnifications were ×200 with each field representing 0.9 × 0.7 mm. Blood vessel density: blood vessels (count red areas)/mm^2^ endometrium stroma (surface light blue area). Lymph vessel density: lymph vessels (count light green areas)/mm^2^ endometrium stroma (surface light blue area).

**Figure 2 ijms-23-08434-f002:**
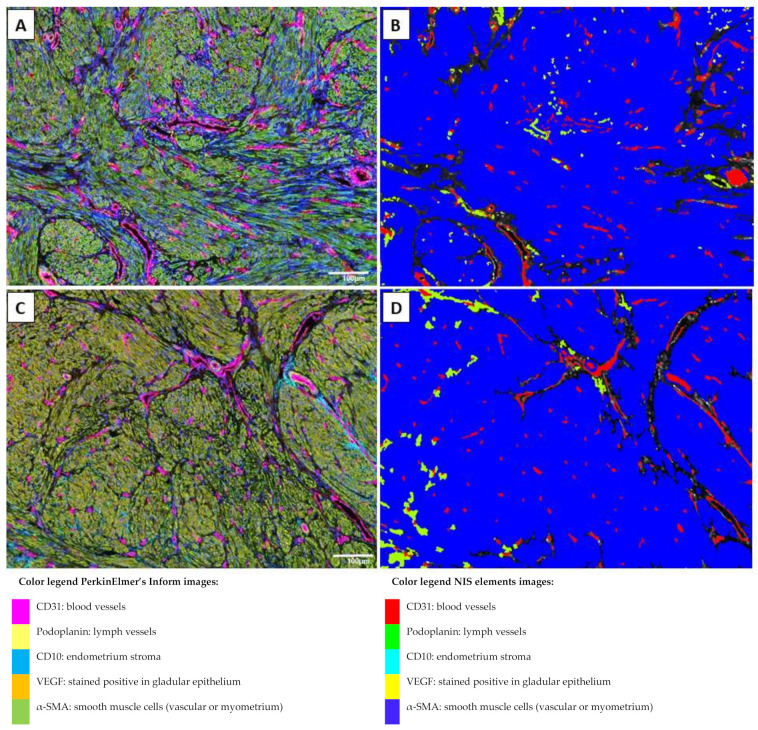
Multiplex images viewed in PerkinElmer’s inForm^®^ advanced image analysis software (**A**,**C**) and analysis of multiplex images in NIS Elements using binary layers for an automated analysis (**B**,**D**) of control myometrium (**A**,**B**), and adenomyosis myometrium (**C**,**D**). All magnifications were ×200 with each field representing 0.9 × 0.7 mm.

**Figure 3 ijms-23-08434-f003:**
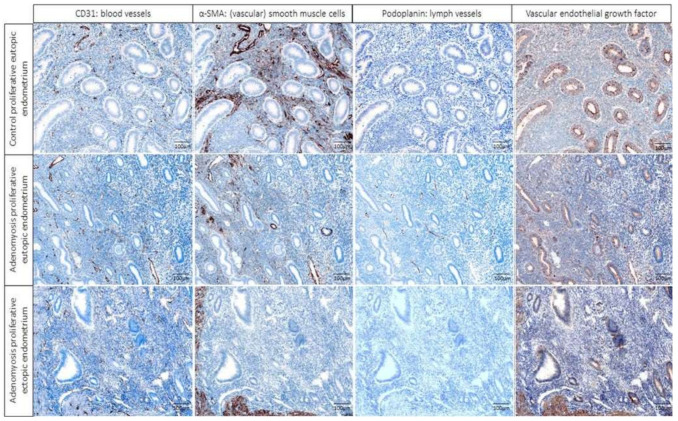
Representative images of the individual IHC markers using the pathology view in PerkinElmer’s inForm^®^ advanced image analysis software, with the primary antibody staining brown (columns from left to right of CD31, a-SMA, podoplanin, and vascular endothelial growth factor (VEGF)) and cell nuclei blue in samples of control proliferative eutopic endometrium (**top row**), adenomyosis proliferative eutopic endometrium (**middle row**), and adenomyosis ectopic endometrium (**bottom row**).

**Figure 4 ijms-23-08434-f004:**
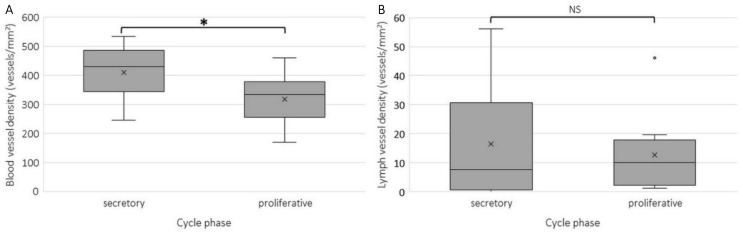
Boxplots of blood vessel density determined by the count of CD31 stained areas (vessels) per mm^2^ CD10 stained area (**A**) and lymph vessel density determined by the count of podoplanin stained areas per mm^2^ CD10 stained area (**B**) in the eutopic endometrium in the secretory (*n* = 20, missing *n* = 4) and proliferative (*n* = 11, missing *n* = 3) menstrual cycle phase. The blood vessel density was significantly higher in the secretory than in the proliferative cycle phase (*p* < 0.05). There was no significant difference in lymph vessel density between the secretory and proliferative cycle phase. Data are shown as median ± range for all patient data pooled. NS: not significant; * significance *p* < 0.05.

**Figure 5 ijms-23-08434-f005:**
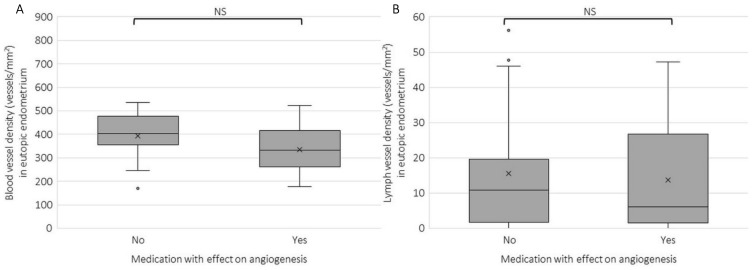
Boxplots of blood vessel density in the eutopic endometrium determined by the count of CD31 stained areas (vessels) per mm^2^ CD10 stained area (**A**) and lymph vessel density determined by the count of podoplanin stained areas per mm^2^ CD10 stained area (**B**) in the eutopic endometrium per medication use, with ‘no’ is no/other medication, and yes is NSAIDs/tranexamix acid. There were no significant differences in blood or lymph vessel density between patients who reported to use medication with an effect on angiogenesis. Data are shown as median ± range for all patient data pooled. NS: not significant.

**Figure 6 ijms-23-08434-f006:**
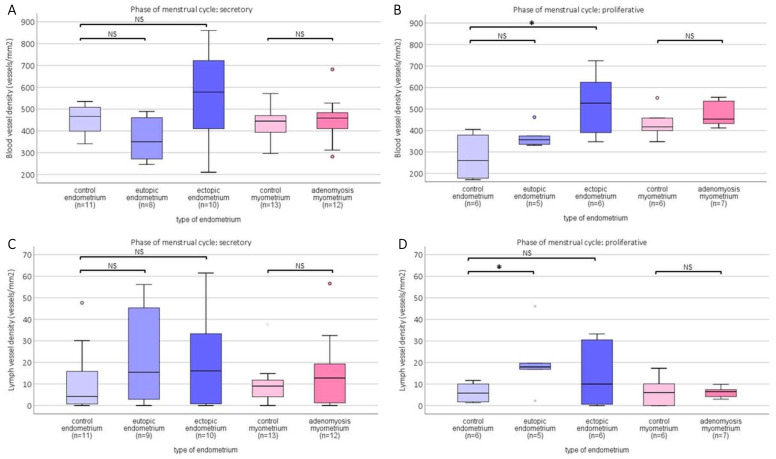
Top row: Boxplots of blood vessel density (BVD) determined by the count of CD31 stained area (vessels) per mm^2^ in the secretory phase (**A**) and proliferative phase (**B**) of the menstrual cycle per tissue type. In the secretory phase, there were no significant differences in BVD between tissue types. In the proliferative phase, the BVD was significantly higher in the ectopic endometrium than in the control endometrium (*p* < 0.05). Bottom row: Boxplots of lymph vessel density (LVD) determined by the count of podoplanin stained area (vessels) per mm^2^ in the secretory phase (**C**) and proliferative phase (**D**) of the menstrual cycle per tissue type. In the secretory phase, there were no significant differences in LVD between tissue types. In the proliferative phase, the LVD was significantly higher in the eutopic endometrium than in the control endometrium (*p* < 0.05). Data shown as median ± range. NS: not significant; * significance *p* < 0.05.

**Figure 7 ijms-23-08434-f007:**
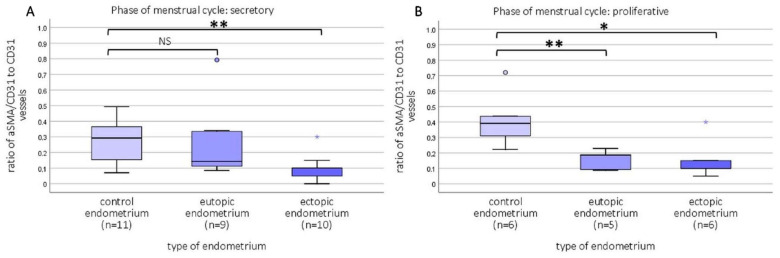
Boxplots of the ratio of the relative number of immature blood vessels, determined by the ratio of mature blood vessels that presented double immunostaining of CD31 and α-SMA to the total number blood vessels in the endometrium in the stroma in the secretory (**A**) and proliferative phase (**B**) of the menstrual cycle. In the secretory phase, the ratio of aSMA/CD31 vessels to CD31 vessels was lower in the ectopic endometrium than in the control endometrium (*p* < 0.01). In the proliferative phase, the ratio of aSMA/CD31 vessels to CD31 vessels was lower compared to the control endometrium in both the eutopic (*p* < 0.01) as well as the ectopic endometrium (*p* < 0.05). Data shown as median ± range. NS: not significant; * significance *p* < 0.05; ** significance *p* < 0.01.

**Figure 8 ijms-23-08434-f008:**
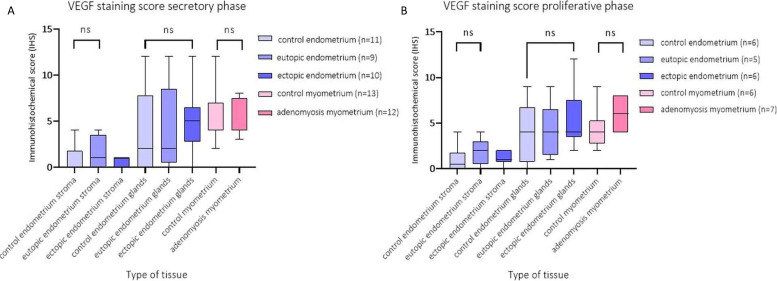
Boxplots of immunohistochemical score (IHS) of VEGF staining per tissue type in the secretory (**A**) and proliferative phase (**B**) in adenomyosis eutopic and ectopic endometrium stroma and glands (*n* = 16) versus control endometrium stroma and glands (*n* = 18), and in adenomyosis myometrium (*n* = 19) versus control myometrium (*n* = 18) (right). There were no significant differences in VEGF IHS in each tissue type between the adenomyosis and control patients. Data shown as median ± range. NS: not significant.

**Figure 9 ijms-23-08434-f009:**
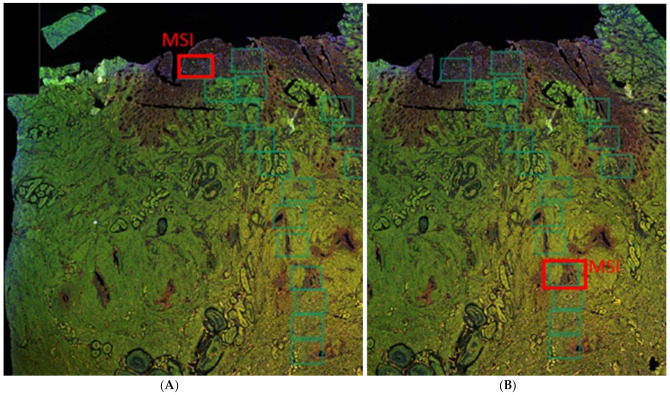
Examples of Phenochart™ whole slide image viewer and the selection of multispectral image (MSI) areas of interest (green rectangles) of eutopic endometrium (red rectangle on panel (**A**)) and ectopic endometrium (red rectangle on panel (**B**)) after multicolor IHC staining of an adenomyosis sample.

**Table 1 ijms-23-08434-t001:** Characteristics of adenomyosis patients and controls.

Item	Patients without Adenomyosis (*n* = 19)	Patients with Adenomyosis (*n* = 19)	Statistical Significance
Median age (range)	43.4 (28–48)	45.4 (32–50)	0.370
Reason for hysterectomy (symptoms)			
Prolapse	11 (57.9%)	0 (0%)	
Dysmenorrhea	3 (15.8%)	3 (15.8%)	
Cervical dysplasia	3 (15.8%)	1 (5.3%)	
Ovarian Cyst	1 (5.3%)	0 (0%)	
Abnormal uterine bleeding	1 (5.3%)	15 (78.9%)	
Menstrual phase			
Proliferative	6 (31.6%)	7 (36.9%)	0.732
Secretory	13 (68.5%)	12 (63.2%)	
Number of patients with representative material			
Eutopic endometrium	17	14	
Proliferative phase	6	5	
Secretory phase	11	9	
Ectopic endometrium	-	16	
Proliferative phase		6	
Secretory phase		10	
Myometrium	19	19	
Proliferative phase	6	7	
Secretory phase	13	12	
Parity			0.118
0	0 (0%)	2 (10.5%)	
1	6 (31.6%)	4 (21.1%)	
2	8 (42.1%)	12 (63.2%)	
3	5 (26.3%)	1 (5.3%)	
Previous cesarean section			0.001
No	19 (100%)	9 (47.4%)	
Yes		10 (52.7%)	
Medication			0.119
None	14 (73.7%)	11 (57.9%)	
Tranexamic acid	0 (0%)	1 (5.3%)	
NSAID (when needed)	5 (26.3%)	3 (15.8%)	
Other *	0 (0%)	4 (21.1%)	

Abbreviations: NSAID, nonsteroidal anti-inflammatory drug. * other medication: iron supplements, levothyroxine sodium, acetaminophen when needed, insulin, and lactulose.

## Data Availability

The data presented in this study are available on request from the corresponding author. The data are not publicly available, due to privacy concerns of the patient material.

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
