# Peer review of "Increased Angiogenesis and Lymphangiogenesis in Adenomyosis Visualized by Multiplex Immunohistochemistry"

_ijms, 2022, doi:10.3390/ijms23158434_

Round 1

Reviewer 1 Report

I would like to acknowledge M. J. Harmsen and the authors for their response to the first cycle of submission and for working out the points addressed in the earlier review. Research articles are thought to be progress reports, and we have learnt from all mentors that even negative results are results. Science is a cumulative, progressive, and linear philosophical approach to knowledge that scientists have thrown through their research throughout history. I reckon, experimentally and scientifically, the authors well-represented the experimental design from variables definition, testable hypothesis, measurements, and statistical testing within the clinical limitation and portrayed well-described inputs, outputs, and constraints.

The authors highlighted the main objective of the MS in the last section of the introduction ”to investigate whether there is evidence for increased  blood and lymph vessel density, level of VEGF expression and relative number of immature blood vessels as markers for new blood vessel formation, in the eutopic- or ectopic endometrium, or in the myometrium of patients with adenomyosis compared to the endometrium or myometrium in controls with unrelated pathologies.”

The methods are immensely formulated to fit and answer the proposed research question. The results demonstrated that blood vessel density was highest in the ectopic endometrium and increased compared to both the eutopic endometrium of patients with and 213 without adenomyosis. The eutopic endometrium of adenomyosis patients had a higher 214 relative number of immature blood vessels, demonstrated by the significantly lower ratio 215 of mature vessels to the total number of blood vessels compared to the control endometrium. In addition, VEGF expression in ectopic/eutopic endometrium was surveyed in the literature, and the current findings were presented as reflected from the statistical testing of the studied samples. The comprehensive discussion, limitations and future implications encourage the acceptance of the manuscript for publication.

Minors to be considered

Final proofreading will be needed for typos and punctuations such as:

-          L24 missed space in (lymph)angiogenesis.

-          L19 missed article= the “under influence of” > under the influence of.

-          Missed hyphen L311, L314 (break through>break-through or breakthrough, and thin walled> thin-walled).

-          Missing comma after ALSO L345.

Others

-          Figures 2 and 3 locations on pages 5 and 6 should be fixed. Figure 3 resolution may be improved.

-          Please indicate the statistical test used to generate P values with the correction, if any, like one-way ANOVA with Tukey corrections of multiple comparisons..etc. (Figures 6 and 7).

-          Fix the x-axis range for figure 8 to display the cropped section of the negative boxplot part of the first three and one boxplots for the left and right subfigures, respectively.

-          It is advised to label the subfigures like A, B, and C if possible.

-          L307 singular verb supports does not appear to agree with the plural subject patients.

-          Some exceptionally long statements could be broken into shorter lines for better reader engagement.

Reviewer 2 Report

In the submitted manuscript by Harmsen and colleagues, uterine tissue sections from patients with adenomyosis are immunolabeled to examine blood and lymphatic vessel densities to determine if differences may be part of the disease etiology. The overall findings in the work are relatively minimal – not many changes in the measured parameters are noted – yet the conclusions and discussion of the work are immense. Some of the biology/physiology may be incorrectly stated as well.

Discussions of the basic physiology here are not fully correct. Fluid accumulation would not result due to increased lymphatic drainage (line 64). This is stated throughout. While lymphangiogenesis often occurs with inflammation, whether it is indicative of lymphatic dysfunction or not is unclear. Similarly, angiogenesis does not necessarily increase leakage and the relative amount of alphaSMA on blood vessels may not indicate anything. The vessels shown in Figure 1 look to be more arterioles, but bleeding or permeability would be from capillaries, which do not have extensive aSMA. VEGF induces permeability (and the NSAIDs, which the authors discount, reduce it). In total, the basic discussion of what is going on and how the data is interpreted is incorrect.

Other:

In Figures 1 and 2 the mages are hard to make out what is what. They could be labeled better and the authors must do something for the colors to be visualized better. It is unclear what the lower panels are and the images lack scale bars.

The number and density on the IHC (Figure 3) does not appear to match that which is in the multiplex images.

The discussion is longer than the whole rest of the paper and must be trimmed to be more specific to the paper findings as the length and detail or others work almost makes this current report of little impact.

In fig 3, the size of the endometrial glands seems to be reduced in adenomyosis group. Could the authors comment?

The authors find no difference in the lymphatic density between proliferative and secretory phase, but it has been shown that lymphatic density changes with different stages of estrous cycle in rodents. How do the authors justify?

The study used VEGF-A for immunostaining, what is the rationale behind using subtype A when C & D are known to be associated with lymphangiogenesis?  

Secretory phase shows high blood vessel density than proliferative phase. The authors could add explanation for better understanding of the readers.

Figure 6 should be recreated so as to be vertical.

Author Response

This manuscript is a resubmission of an earlier submission. The following is a list of the peer review reports and author responses from that submission.

Round 1

Reviewer 1 Report

Harmsen et al. explored the role of angio/lymphangio-genesis in adenomyosis. The authors utilised patient and control samples and the groups were compared via multiplex IHC. The MS is well written with very minor typos throughout the article. the experimental design is adequate, and the results is fine while a better presentation of statistical significance could be reflected in the figure legends. I enjoyed the discussion and the future direction and delighted to accept its publication after final proof reading check; for example, l24 lymh)angiogenesis> Lymph) angio, l128 and l134 Data shown> data are shown, missing articles inappropriate punctuation all over the article.

Author Response

Response to reviewer 1

We want to thank the reviewer for taking time to review our manuscript and the positive feedback. In response to the reviewer we amended the following points;

  • We checked the manuscript for typos
  • The figure legends were altered to better reflect the statistical significance
    • Figure 4
    • Figure 5
    • Figure 6
    • Figure 7
    • Figure 8
  • We checked the manuscript for punctuation errors

Reviewer 2 Report

Sir, 

I have reviewed recently the manuscript "Increased angiogenesis and lymphangiogenesis in adenomyosis visualized by multiplex immunohistochemistry" submitted by Marissa J. Harmsen and co-workers to IJMS. 

The authors aimed to study morphologically adenomyosis -  a benign uterine condition associated with abnormal uterine bleeding and subfertility. 

The authors present their data acquired by multiplex immunohistochemistry. This seems to be a very robust technology and an interesting research tool, indeed. However,  I am not much convinced by the whole study design. After careful reading of the manuscript, I am sorry to say that the manuscript leaves an impression that the authors decided to use the technology and the histology/biology behind are somewhat immaterial. 

The study is based on very conventional markers (aSMA, podoplanin, CD31...). The overall design is purely descriptive. The authors aimed to perform this study because " it is vital to gain insight into morphology and function of adenomyosis tissue". Well,  this is not much experimental approach and it is also a very weak working hypothesis. 

The authors should clarify what could be the practical outcome of their study. Do these findings have any practical application? I believe that diagnostics does not require this sort of IHC analysis for confirmation of adenomyosis. 

The authors have listed very well various limitations of their study (section 3.1). I must entirely agree and I appreciate their honesty.  However, there is very little to highlight as a relevant result. Mostly, the results presented do not reach statistical significance. Also, the data does not provide insight into the function of adenomyosis tissue which was the primary objective. 

As it is, the manuscript does not contain sufficiently sound novel data to justify its publication. I must regrettably suggest the rejection of the manuscript. 

Author Response

Response to reviewer 2 (also in attachment):

While we used conventional markers, the multiplex technology allowed the simultaneous analysis of these markers in one image. The greatest advantage was that we could perform the vessel counts very accurately per tissue type, examples of which are presented in Figure 1 and 2.

We agree with the reviewer that the study is rather descriptive. Nevertheless, although we state in line 41-44 that it is vital to gain insight into the morphology and function of adenomyosis tissue, this is not the working hypothesis. Instead, in line 66-71 we state that the main objective of this study was to “investigate whether there is evidence for increased blood and lymph vessel density, level of VEGF expression and relative number of immature blood vessels as markers for new blood vessel formation, in the eutopic- or ectopic endometrium, or in the myometrium of patients with adenomyosis compared to the endometrium or myometrium in controls with unrelated pathologies.” We added ‘more’ to line 42 to indicate that we only analyze a small part of this overall aim.

These objectives were analyzed through vessel counts and staining scores, which are conventional methods in immunohistochemistry. We believe we performed the analyses in the most optimal way, and we list the strengths and limitations of this method in the discussion, section 3.1.

confirmation of adenomyosis. 

We agree with the reviewer it would be good to mention the practical outcome of this study. We also agree with the reviewer that multiplex IHC is not needed to confirm the presence of adenomyosis. Rather, we think that the results should be regarded as enhanced insight into the biology of adenomyosis, mainly the role of vascular biology, which may deliver indicators to perform further research to investigate whether the abnormal bleeding and intracavitary fluid in adenomyosis patients is associated with angiogenesis and lymphangiogenesis, as mentioned in line 348-352 in the discussion.

We added line 355-358 to clarify: Although the assessment of blood- or lymph vessel density through the use of multiplex immunohistochemistry, as was performed in the current study, does not aid in the diagnosis of adenomyosis, it might be helpful to enhance insight into the importance of angiogenesis and Lymphangiogenesis and determine the clinical relevance or severity of the adenomyosis.    

The small sample size and the retrospective character of this study limit the relevance of our results. However, we also believe that the detailed methodology of the multiplex IHC applied, and the difficulty to select good patient and control sample of this population justifies the publication of our results. Furthermore, we explained above that the objective is listed in a more detailed manner in the last paragraph of the introduction. We deleted ‘function’ from line 43, since we agree with the reviewer that investigating the function of adenomyosis tissue reaches beyond the scope of this study.
